# Meta-Analysis of Dietary Curcumin Supplementation in Broiler Chickens: Growth Performance, Antioxidant Status, Intestinal Morphology, and Meat Quality

**DOI:** 10.3390/antiox14040460

**Published:** 2025-04-12

**Authors:** Pedro Abel Hernández-García, Lorenzo Danilo Granados-Rivera, José Felipe Orzuna-Orzuna, Gabriela Vázquez-Silva, Cesar Díaz-Galván, Pablo Benjamín Razo-Ortíz

**Affiliations:** 1Centro Universitario Amecameca, Universidad Autónoma del Estado de México, Amecameca CP 56900, Mexico; pedro_abel@yahoo.com (P.A.H.-G.); mvzrazo@gmail.com (P.B.R.-O.); 2Campo Experimental General Terán, Instituto Nacional de Investigaciones Forestales, Agrícolas y Pecuarias, General Terán CP 67400, Nuevo León, Mexico; granados.danilo@inifap.gob.mx; 3Departamento de Zootecnia, Universidad Autónoma Chapingo, Chapingo CP 56230, State of Mexico, Mexico; 4Departamento del Hombre y su Ambiente, Universidad Autónoma Metropolitana—Xochimilco, Mexico City CP 04960, Mexico; gavaz@correo.xoc.uam.mx; 5Departamento de Producción Agrícola y Animal, Universidad Autónoma Metropolitana—Xochimilco, Mexico City CP 04960, Mexico; cesarwardi14@gmail.com

**Keywords:** polyphenolic compound, growth promoter, carcass yield, meta-regression

## Abstract

This study aimed to evaluate the effects of dietary curcumin supplementation on growth performance, serum antioxidant status, intestinal morphology, and meat quality of broiler chickens. The database was constructed with 28 peer-reviewed scientific papers published between January 2013 and January 2025, identified and selected from the Scopus, Web of Science, ScienceDirect, and PubMed databases following the PRISMA methodology. All response variables in the database were analyzed with random effects models using the R statistical software, and the results were reported as weighted mean differences (WMD). Dietary curcumin supplementation did not affect (*p* > 0.05) average daily feed intake. However, dietary curcumin supplementation increased (*p* < 0.001) daily weight gain and decreased (*p* < 0.001) feed conversion ratio. Dietary curcumin supplementation increased (*p* < 0.01) serum superoxide dismutase, catalase, glutathione peroxidase, and total antioxidant capacity but decreased (*p* < 0.001) serum malondialdehyde (MDA) concentration. Dietary curcumin supplementation decreased (*p* < 0.001) crypt depth (CD) and increased (*p* < 0.001) villus height (VH) and VH/CD ratio in the duodenum. Dietary curcumin supplementation increased (*p* < 0.05) carcass yield and color (L*, a*, and b*) in meat and, at the same time, decreased (*p* < 0.01) cooking loss and fat and MDA content in meat. Curcumin can be used as a dietary additive to improve productive performance, antioxidant status in blood serum, intestinal morphology, and meat quality in broiler chickens.

## 1. Introduction

According to Uzundumlu and Dilli [1], the global demand for chicken meat will increase by around 3.0% in the next 6 years starting in 2025. To meet this demand for chicken meat, it is necessary to improve the productive performance of broiler chickens [2]. However, broiler chickens are exposed to various diseases, which can negatively affect their productivity and cause farm economic losses [3]. Ogbuewu et al. [4] mention that the inclusion of antibiotics in broiler feed has been used for several decades as a nutritional strategy to prevent and treat diseases and stimulate animal growth. However, the misuse of antibiotics in broiler diets has led to toxic residues in meat and bacteria resistant to their effects, which is a significant risk for human health worldwide [5]. Consequently, poultry researchers are looking for new natural products that can improve broiler chickens’ productive performance and health without the adverse effects of antibiotics [6,7]. According to several recent studies [3,6,8], some secondary metabolites extracted from plants, such as tannins, dihydroartemisinin, and curcumin, have shown promising effects in improving productive performance and health status in broiler chickens, without using antibiotics.

Curcumin is a hydrophobic polyphenolic compound that is extracted from the rhizome of turmeric (*Curcuma* spp.) and has antioxidant, anti-inflammatory, immunomodulatory, and antimicrobial effects in poultry [5,8,9]. Particularly in broiler chickens, several scientific articles have been published in recent years testing the effects of dietary supplementation with curcumin on productive performance [10,11], antioxidant status in the blood [12,13], small intestine morphology [7,14], and meat quality [15,16]. However, inconsistent results have been observed among some of these scientific articles, which limits the appropriate recommendation of curcumin as a dietary additive for broiler chickens. For example, Xu et al. [12] and Salah et al. [10] observed positive effects on productive performance and antioxidant status in the blood serum of broilers supplemented with low doses of curcumin (10 to 100 mg/kg feed). Similarly, Fathi et al. [14] and Gumus et al. [15] observed positive effects on intestinal morphology and meat quality of broilers supplemented with low to moderate doses (25 to 500 mg/kg feed) of curcumin. However, other studies [11,13] reported that dietary supplementation with increasing doses (between 200 and 2000 mg/kg feed) of curcumin did not affect productive performance and antioxidant status in broilers. Likewise, some authors [7,16] reported that dietary supplementation with curcumin (100 to 800 mg/kg of feed) did not affect intestinal morphology and meat quality in broiler chickens. According to several researchers [8,9,17], the dose of curcumin included in the diets, the type of curcumin (standard or nanoparticles), and the period of curcumin supplementation are factors that influence the variability of the effects of curcumin as a dietary additive for broiler chickens.

Several recent narrative reviews [5,8,9,17] have suggested that dietary supplementation with curcumin has a positive impact on the productive performance, oxidative status, intestinal morphology, and meat quality of poultry. However, traditional literature reviews are descriptive and lack methodological rigor since, according to Tawfik et al. [18], in these reviews, the authors generally select the articles based on their own opinion. In contrast, Paul and Barari [19] indicate that the meta-analytic methodology is a research method that systematically combines and synthesizes data extracted from multiple studies, improving the conclusions’ reliability. A recent meta-analysis [4] evaluated the effects of the *Curcuma longa* plant (oil and powder) as a feed additive for broiler chickens. However, based on the literature reviewed during the current meta-analysis’s development, no previous scientific articles have used the meta-analytic methodology to evaluate the effects of curcumin as a dietary additive for broiler chickens or other poultry. Compared to the *C. longa* plant, curcumin is generally required in lower amounts (1 g/kg vs. 5 g/kg) and has higher antioxidant and growth-promoting potential [4,5,8], which makes it more suitable for evaluation as an additive for broiler chickens. Furthermore, the effects of the *C. longa* plant cannot be attributed to a specific nutrient or secondary metabolite due to its broad content of chemical compounds [4,8]. The current study’s hypothesis states that using curcumin as a dietary additive for broiler chickens will benefit the animals’ productive performance, blood serum oxidative status, intestinal morphology, and meat quality. Therefore, the objective of the current study was to evaluate the effects of dietary curcumin supplementation on productive performance, blood serum antioxidant status, intestinal morphology, and meat quality of broiler chickens using the meta-analytic methodology.

## 2. Materials and Methods

### 2.1. Literature Search

The research question of the current meta-analysis was formulated following the Population (P), Intervention (I), Comparison (C), and Outcomes (O) format proposed by Nishikawa-Pacher [20]. Specifically, P was broiler chickens, I was dietary supplementation with curcumin, C was between control diets (without added curcumin) and diets containing curcumin, and O were the values reported in the response variables of productive performance, antioxidant status in blood serum, intestinal morphology, and meat quality. From the identification to the inclusion of scientific articles that evaluated the effects of curcumin as a dietary additive for broiler chickens, the PRISMA guidelines described by Moher et al. [21] were followed, as shown in Figure 1 and PRISMA checklist (Appendix A). Eligible scientific articles were identified with systematic searches restricted to information published between January 2013 and January 2025 (to obtain and display updated information), in English and available in the electronic databases PubMed, Google Scholar, ScienceDirect, and Scopus. In all these databases, the following keywords were used: (1) broiler chickens, (2) curcumin, (3) productive performance, (4) antioxidant status, (5) intestinal morphology, and (6) meat quality.

### 2.2. Exclusion and Inclusion Criteria

The exclusion criteria for the current meta-analysis were the following: (1) book chapters, conference proceedings, review articles, and university theses; (2) studies that used curcumin combined with other plant secondary metabolites, or antimicrobials; and (3) studies that used broiler chickens experimentally infected with protozoa, viruses, bacteria or aflatoxins in the diet. Some studies separately evaluated the effects of curcumin in clinically healthy and experimentally infected animals. From these studies, only data from clinically healthy animals were included in the current meta-analysis database. On the other hand, the inclusion criteria were the following: (1) experiments that used clinically healthy broiler chickens not challenged with protozoa, bacteria, or viruses as experimental model; (2) scientific articles published between January 2013 and January 2025 in English language and peer-reviewed journals; (3) scientific articles that reported treatment means, number of replicates (*n*) and standard error (SEM) or standard deviation (SD) in at least one response variable related to productive performance, antioxidant status in blood serum, intestinal morphology in the duodenum, or meat quality; (4) studies that compared at least one experimental group supplemented with curcumin through the diet against a control group not supplemented with curcumin in the diet; and (5) studies that reported the supplementation period and the doses of curcumin (mg/kg of feed) used.

### 2.3. Data Extraction

The extraction format reported by other authors [2,22] was used to extract and sort the data on the first author’s surname, year of publication, breed of broilers used, days of experiment, dose of curcumin (mg/kg of feed) added to the diets, and type of curcumin used (standard curcumin or nanoparticles) from the 28 scientific articles included in the current meta-analysis database (Table 1). From each of the articles included in Table 1 the mean, SD or SEM, and *n* were extracted for the following groups of response variables: Group 1, average daily feed intake (ADFI), daily weight gain (DWG), and feed conversion ratio (FCR). Group 2. serum concentration of malondialdehyde (MDA), total antioxidant capacity (T-AOC), glutathione peroxidase (GSH-Px), catalase (CAT), and superoxide dismutase (SOD). Group 3, villus height (VH), crypt depth (CD), and VH/CD ratio in the duodenum. Group 4, carcass yield, and fat, protein and MDA content in meat, shear force (ShF), cooking loss (CL), lightness (L*), redness (a*), yellowness (b*), and pH of breast meat. The means, SEM, and SD reported in graphs were extracted following the procedures described by Drevon et al. [23] using version 4.4 of the WebPlotDigitizer software. As recommended by Lucio-Ruíz et al. [24] and Orzuna-Orzuna and Granados-Rivera [25], the current meta-analysis only included response variables reported in at least three scientific articles.

### 2.4. Calculations, Statistical Analysis, Heterogeneity, and Publication Bias

All data were analyzed using specialized statistical packages (“meta” and “metafor”) available in version 4.1.2 of the R statistical software, as described in detail by Viechtbauer [46]. Effect sizes were expressed as weighted mean differences (WMD) with 95% confidence intervals (CI), which were calculated using random effects models with the methods proposed by Der-Simonian and Laird [47]. On the other hand, heterogeneity between studies was examined statistically using Cochran’s Q test and the I^2^ statistic [2,22]. When a *p* ≤ 0.05 was obtained in Cochran’s Q test and, at the same time, a value > 50% in the I^2^ statistic, heterogeneity was considered significant [48]. Publication bias was examined using the Begg [49] and Egger [50] statistical tests with a significance level of *p* ≤ 0.05.

### 2.5. Meta-Regression and Subgroup Analysis

Using the Der-Simonian and Laird [47] method of moments, univariate meta-regression analyses were applied to investigate the role of the covariates breed (Arbor Acres, Ross 308, and Cobb 500), supplementation period (≤21 and >21 days), curcumin dose (≤200, 201–500, and >500 mg/kg feed), and curcumin type (standard curcumin or nanoparticles) on the observed heterogeneity in ADFI, DWG, and FCR. Other response variables were not evaluated through meta-regression because they were reported in less than ten scientific articles and, according to Lucio-Ruíz et al. [44] and Yibar and Uzabaci [22], false positives in the results can be obtained under these conditions. As recommended by other studies [2,45], subgroup analyses were applied to evaluate significant (*p* ≤ 0.05) covariates in ADFI, DWG, or FCR.

## 3. Results

### 3.1. Growth Performance

Table 2 shows that ADFI was not affected (*p* > 0.05) by dietary curcumin supplementation. However, ADG increased (*p* < 0.001) in response to dietary curcumin supplementation. In contrast, dietary curcumin supplementation decreased (*p* < 0.001) FCR.

### 3.2. Antioxidant Status in Blood Serum

Table 3 shows that dietary curcumin supplementation increased (*p* < 0.01) the serum concentration of SOD, CAT, GSH-Px, and T-AOC. However, dietary curcumin supplementation decreased (*p* < 0.001) the serum concentration of MDA.

### 3.3. Intestinal Morphology

Dietary curcumin supplementation increased (*p* < 0.001) VH and VH/CD ratio in the duodenum (Table 4). In contrast, dietary curcumin supplementation decreased (*p* < 0.001) CD in the duodenum.

### 3.4. Carcass Yield and Meat Quality

Dietary curcumin supplementation increased (*p* < 0.05) carcass yield as well as a* and b* in meat (Table 5). However, dietary curcumin supplementation did not affect (*p* > 0.05) L*, meat pH, ShF, and protein content. In contrast, dietary curcumin supplementation decreased (*p* < 0.01) CL and fat and MDA content in meat.

### 3.5. Publication Bias and Meta-Regression

The two tests (Egger and Beg) used to assess publication bias were not significant (*p* > 0.05) in any of the response variables evaluated in the current study (Table 2, Table 3, Table 4 and Table 5), indicating the absence of publication bias.

Table 2 shows heterogeneity (*p* < 0.001) in ADFI, DWG, and FCR. Likewise, heterogeneity (*p* < 0.05) was detected in SOD, CAT, and GSH-Px (Table 3), while Table 4 shows that there was heterogeneity (*p* < 0.001) in VH, CD, and VH/CD ratio. In addition, heterogeneity (*p* < 0.001) was detected in carcass yield, meat fat content, and meat MDA content (Table 5). However, Koricheva et al. [48] mention that the statistical accuracy and reliability of meta-regression is low when applied to response variables reported in less than ten studies. Therefore, the current study only applied meta-regression to ADFI, DWG, and FCR.

Table 6 shows that the covariates breed/strain of broilers and type of curcumin (standard or nanoparticles) used were not statistically related (*p* > 0.05) to the response variables evaluated (ADFI, DWG, and FCR). The supplementation period explained (*p* < 0.05) 11.45% and 12.04% of the heterogeneity observed in ADFI and DWG, respectively. Likewise, the curcumin doses explained (*p* < 0.05) 10.79% and 9.16% of the heterogeneity observed in DWG and FCR, respectively.

### 3.6. Subgroup Analysis

Figure 2a shows that ADFI increased (WMD = 3.321 g/d; *p* = 0.005) when short (≤21 days) periods of curcumin supplementation were used. However, ADFI was not affected (*p* > 0.05) when long (>21 days) periods of curcumin supplementation were used. In contrast, DWG increased (WMD = 1.464 g/d; *p* < 0.001) when long (>1 days) periods of curcumin supplementation were used (Figure 2b). However, DWG was not affected (*p* > 0.05) when short (≤21 days) periods of curcumin supplementation were used.

Figure 3a shows that DWG increased (WMD = 2.530 g/d; *p* < 0.001) when low doses (≤200 mg/kg feed) of curcumin were used. However, moderate (201–500 mg/kg feed) and high (>500 mg/kg feed) doses of curcumin did not affect DWG (*p* > 0.05). In contrast, Figure 3b shows that FCR decreased (WMD = 2.530 g/d; *p* < 0.001) when low doses (≤200 mg/kg feed) of curcumin were used. However, FCR was not affected (*p* > 0.05) when moderate (201–500 mg/kg feed) and high (>500 mg/kg feed) doses of curcumin were used.

## 4. Discussion

### 4.1. Growth Performance

According to recent reviews [5,9], curcumin could be a potential nutritional strategy to stimulate poultry growth rate without antibiotics. In the current study, dietary supplementation with curcumin did not affect ADFI, but increased DWG and decreased FCR. These findings indicate that curcumin intake can be used as a natural growth promoter in broiler chickens and, at the same time, improve feed efficiency. Similarly, recent studies [51,52] observed positive effects on DWG and FCR of Japanese quail and ducks supplemented with increasing doses (100, 200, 300, 400, and 500 mg/kg feed) of dietary curcumin. In the current meta-analysis, dietary curcumin supplementation increased VH in the duodenum, indicating a larger area of nutrient absorption that could lead to higher DWG and lower FCR. Furthermore, dietary curcumin supplementation decreased (−30.2 to −45.1%) the oocyst counts of *Eimeria acervulina*, *Eimeria tenella*, and *Eimeria maxima* [37], which are negatively correlated (r = −0.51 to −0.63) with DWG in broilers [53]. On the other hand, Chen et al. [25] reported that dietary curcumin supplementation improves the mRNA expression of sodium-glucose cotransporter 1 (*SGLT1*), glucose transporter 2 (*GLUT2*), peptide transporter 1 (*PEPT1*), and cationic amino acid transporter 1 (*CAT1*) in the small intestine of broiler chickens. These effects of curcumin intake in broiler chickens might be related to the higher DWG and lower FCR observed in the current study since, according to Gilbert et al. [54], *SGLT1* and *GLUT2* facilitate glucose absorption and reabsorption, while *PEPT1* facilitates di/tri peptide absorption in enterocytes, and *CAT1* facilitates the movement of essential cationic amino acids (lysine and arginine) into body cells.

Some subgroup analyses showed that the best response in DWG was obtained with long supplementation periods (>21 days) and low doses (≤200 mg/kg feed) of curcumin, while the best FCR was obtained with low doses (≤200 mg/kg feed) of dietary curcumin. These positive effects of low doses of curcumin for long periods of supplementation could be explained by the findings reported by Rajput et al. [34], who observed that dietary supplementation with low doses (≤200 mg/kg feed) of curcumin increased the digestibility of crude protein, ether extract, and metabolizable energy in broiler chickens, particularly with supplementation periods longer than 21 days.

### 4.2. Antioxidant Status

According to Surai and Fisinin [55,56], broilers in intensive production systems are exposed to nutritional (mainly feed contaminated with mycotoxins), environmental (thermal stress and ammonia accumulation), and physiological (high growth rate) stressors. In poultry, these stressors increase the production of reactive oxygen species (ROS) at the cellular level, which leads to oxidative stress (OS) [57]. In broilers, OS negatively affects weight gain and feed efficiency, causes immunosuppression, and increases mortality [55,56]. In the present study, an increase in serum concentration of CAT, SOD, and GSH-Px, was observed in response to dietary supplementation with curcumin. These increases could positively impact the oxidative status of broiler chickens since, in poultry, the first line of antioxidant defense is formed by SOD, CAT, and GSH-Px [5,58]. Specifically, SOD is involved in the conversion of superoxide ($O_{2}^{-}$) to hydrogen peroxide (H_2_O_2_) [56]. Furthermore, Oke et al. [59] indicate that CAT and GSH-Px can neutralize H_2_O_2_ molecules into water and molecular oxygen. Any decrease in the activity of SOD, CAT, and GSH-Px can lead to the accumulation of $O_{2}^{-}$ and H_2_O_2_ in the body, which increases the risk of OS in broiler chickens [58]. Similarly to the results of the current study, Reda et al. [51] and Ruan et al. [52] also reported positive effects of dietary curcumin supplementation on serum levels of GSH-Px, CAT, and SOD in Japanese quail and ducks.

Recent studies [36,42] reported that dietary supplementation with curcumin increases (11.9 to 61.1%) the mRNA of genes encoding SOD, CAT, and GSH-Px in various body tissues of broilers. Therefore, similar effects of curcumin intake in the current meta-analysis might be directly related to the increased serum levels of SOD, CAT, and GSH-Px. Furthermore, curcumin could increase serum levels of SOD, CAT, and GSH-Px indirectly since, in broiler chickens, curcumin increases Nrf2 (nuclear factor erythroid 2-related factor 2) mRNA by up to 22.6% [42], which participates in the activation of SOD, CAT, and GSH-Px synthesis [59].

According to some authors [12,57], serum T-AOC values include enzymatic antioxidants (e.g., SOD, CAT, and GSH-Px) and non-enzymatic antioxidants, such as those supplied through the diet (e.g., vitamin E, carotenoids, and polyphenols). Although curcumin is a polyphenol with antioxidant properties, in poultry and other non-ruminant animals, orally administered curcumin has low intestinal absorption and is rapidly metabolized and excreted from the body [5,60]. Therefore, the higher serum T-AOC concentration observed in the current meta-analysis could be mainly explained by the observed increase in SOD, CAT, and GSH-Px and not by direct antioxidant effects of curcumin. On the other hand, MDA is one of the different chemical compounds formed due to lipid peroxidation [61] and has been widely used worldwide as a biomarker of lipid peroxidation in poultry blood serum [57]. The lower serum concentration of MDA observed in the current study is positive since, according to Ncho et al. [58], a high serum concentration of MDA indicates high ROS-induced damage to phospholipids in cell membranes. Furthermore, serum MDA levels are negatively correlated (r = −0.77 to −0.81) with DWG and feed efficiency in poultry [62]. Therefore, low serum MDA levels could benefit productive performance in broilers.

### 4.3. Intestinal Morphology

According to Fathi et al. [14], it is important to measure intestinal morphology to assess the integrity and nutrient absorption capacity in the small intestine of broiler chickens. VH and CD values are widely used parameters to assess intestinal morphology in poultry [63]. In the current meta-analysis, dietary curcumin supplementation decreased CD and simultaneously increased VH and VH/CD ratio in the duodenum. Similarly, other authors [63,64] also observed an increase in VH and VH/CD ratio and, at the same time, a reduction in CD in the duodenum of Japanese quail and laying hens supplemented with various doses (100 to 500 mg/kg feed) of curcumin. In broiler chickens, an increase in VH indicates a greater surface area for nutrient absorption per unit of intestinal area [14]. Furthermore, the observed reduction in CD suggests that curcumin decreases cell turnover of intestinal villi damaged by pathogen toxins, inflammation, or tissue shedding in broiler chickens [7,34]. High VH/CD values in poultry indicate better intestinal epithelium integrity and health [13].

The positive effects of curcumin on VH, CD, and VH/CD ratio observed in the current study might be related to curcumin’s antimicrobial and antioxidant properties in poultry. For example, in the digestive tract of broiler chickens, dietary supplementation with increasing doses (50 to 400 mg/kg feed) of curcumin decreases (−11.5 to −27.6%) the presence of *Escherichia coli* bacteria [14,32], which produce and release toxins that damage intestinal epithelial cells in poultry [51]. Likewise, Guo et al. [13] observed increased (10.4 to 97.4%) activity of GSH-Px, CAT, and SOD in the duodenal epithelium of broiler chickens supplemented with curcumin (200 mg/kg feed), which could positively impact VH, CD, and VH/CD ratio by decreasing oxidative damage in duodenal tissue. Furthermore, in the small intestines of broiler chickens, dietary supplementation with curcumin (200 mg/kg feed) increases (25.5 to 40.0%) the mRNA expression levels of claudin 1 and zonula occludens-1 [13,25], which are tight junction proteins that play an important role in maintaining good health and integrity in the intestinal epithelium of broiler chickens [63].

### 4.4. Carcass Yield and Meat Quality

The carcass yield of broiler chickens is an important parameter in the meat industry and is related to the economic profitability of farms [65]. On the other hand, Gumus et al. [15] indicate that the color of chicken meat is one of the main parameters that directly influence consumers’ choice and purchase of fresh meat. Likewise, other meat parameters, such as pH, water holding capacity (WHC), ShF, nutritional content, and oxidative stability, influence broiler meat quality [66]. In the current study, dietary supplementation with curcumin improved carcass yield, which could benefit the amount of meat produced and economic profitability in broiler farms. Likewise, in the present meta-analysis, dietary curcumin supplementation did not affect meat pH and L* but increased a* and b*. In chicken meat, pH is an important parameter because it influences the color and water-holding capacity of meat [15], while L* values serve as an indicator to detect the appearance of PSE (pale, soft, and exudative) meat [66]. The antioxidant properties of curcumin may increase a* in meat by decreasing the oxidation of myoglobin to metmyoglobin, just as other antioxidant plant polyphenols act to increase a* values in meat [67]. According to Zhang et al. [68], high a* values could have a low impact on chicken meat quality because red color is not a highly desirable parameter in chicken and other poultry meat. On the other hand, b* values in chicken meat are as important as a* values in ruminant meat since, according to Hayat et al. [69], some consumer populations associate b* with greater freshness and quality in chicken meat. Curcumin is responsible for turmeric’s yellow color [5] and has shown potential as a natural yellow pigment in broiler meat and skin [33]. This would explain the higher b* observed in meat from chickens supplemented with curcumin in the current study.

In the present meta-analysis, dietary supplementation with curcumin decreased CL without affecting ShF, suggesting that curcumin improves the water-holding capacity of chicken meat during cooking without affecting its tenderness [7]. In poultry meat, lower CL increases sensory juiciness in meat and its preference by consumers [70]. Dietary supplementation with curcumin decreases protein carbonylation in the muscle tissue of broiler chickens [7,39]. This effect could explain the lower CL observed in the current meta-analysis since, according to Estévez [71], CL in poultry meat decreases when protein carbonylation in muscle tissue is low.

The protein and fat values in meat observed in the current study were within the range reported as normal by Mir et al. [66] for protein (18.4 to 23.4%) and fat (1.1 to 6.0%) in chicken meat. However, although protein content was not affected in the present study, fat content was decreased in meat from broilers supplemented with dietary curcumin. Recent studies [11,36] reported that dietary supplementation with increasing doses (500, 1000, and 2000 mg/kg feed) of curcumin decreases the mRNA expression of peroxisome proliferator-activated receptor γ (*PPARγ*) and fatty acid synthase (*FASN*) in the muscle tissue of broiler chickens. This mechanism of action of curcumin could explain the lower fat content in chicken meat observed in the current study since, according to a previous study [72], there is a positive correlation (r between 0.55 and 0.60) between meat fat content and *PPARγ* and *FASN* mRNA expression levels in muscle tissue of broiler chickens.

According to Faustman et al. [73], MDA in meat is an aldehyde-type chemical produced by the oxidation of unsaturated fatty acids. The reduction in MDA content in broiler meat observed in the current study suggests that curcumin may improve shelf life in chicken meat since, according to a recent study [59], high MDA content in poultry meat causes off-flavors and discoloration in the meat. Some recent experiments [7,10,30] show that dietary supplementation with curcumin (100 to 1000 mg/kg feed) decreases ROS by up to 22.2% and increases antioxidant enzyme activity (GSH-Px, CAT, and SOD) by up to 40.3% in muscle tissues of broiler chickens. These previously reported mechanisms of action suggest that curcumin could decrease MDA content in broiler chicken meat by decreasing lipid oxidation.

## 5. Conclusions

Curcumin can be used as a dietary additive to improve daily weight gain and decrease the feed conversion ratio in broiler chickens without altering average daily feed intake. The best results for daily weight gain are obtained with low doses (≤200 mg/kg feed) of curcumin and long (>21 days) supplementation periods. Similarly, the best feed conversion ratio is obtained with low doses (≤200 mg/kg feed) of curcumin in diets. Furthermore, dietary supplementation with curcumin improves intestinal morphology in the duodenum and antioxidant status in the blood serum of broiler chickens. Likewise, the use of curcumin as a dietary additive for broiler chickens improves carcass yield, color, and water-holding capacity in meat while simultaneously decreasing fat content and lipid peroxidation in meat.

## Figures and Tables

**Figure 1 antioxidants-14-00460-f001:**
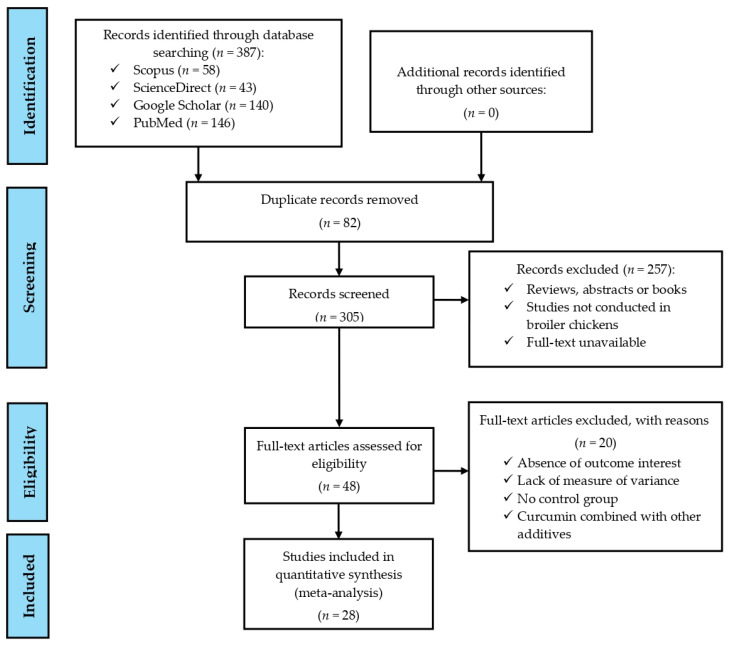
A PRISMA flow diagram detailing the literature search strategy and study selection for the meta-analysis.

**Figure 2 antioxidants-14-00460-f002:**
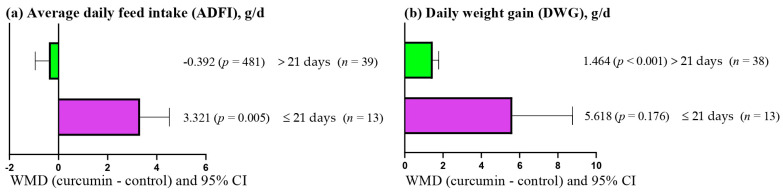
Subgroup analysis (subgroup = supplementation period) of the effect of including curcumin in broiler diets, WMD = weighted mean differences between curcumin treatments and control.

**Figure 3 antioxidants-14-00460-f003:**
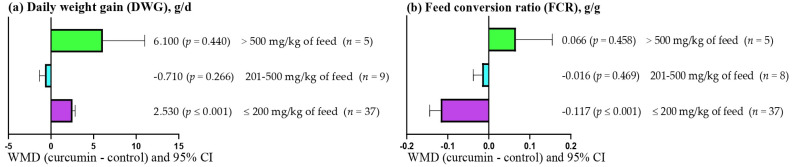
Subgroup analysis (subgroup = curcumin dose (mg/kg of feed)) of the effect of including curcumin in broiler diets, WMD = weighted mean differences between curcumin treatments and control.

**Table 1 antioxidants-14-00460-t001:** Description of the studies included in the meta-analysis database.

Reference	Breed	Supplementation Period	Dose (mg/kg Feed)	Curcumin Type
Abd El-Hack et al. [26]	Not reported	21, 34	5000	Nanoparticles
Badran et al. [27]	Ross 308	35	25, 50, 100	Standard, nanoparticles
Chen et al. [28]	Not reported	52	200	Standard
Eleiwa et al. [29]	Ross 308	42	450	Standard
Fathi et al. [14]	Ross 308	42	25, 50, 100, 200	Nanoparticles
Galli et al. [7]	Ross 308	42	100	Standard
Galli et al. [30]	Cobb 500	44	50	Standard
Gumus et al. [15]	Ross 308	42	250, 500	Standard
Gümüş et al. [31]	Ross 308	42	250, 500	Standard
Guo et al. [13]	Ross 308	28	200	Standard
Hafez et al. [32]	Cobb 500	42	100, 200	Standard
Johannah et al. [33]	Not reported	42	500, 1000	Standard
Pornanek y Phoemchalard [16]	Arbor Acres	42	200, 400, 600, 800	Standard
Rahmani et al. [34]	Ross 308	42	200, 400	Standard, nanoparticles
Rahmani et al. [35]	Ross 308	42	200, 400	Standard, nanoparticles
Rajput et al. [36]	Arbor Acres	42	100, 150, 200	Standard
Rajput et al. [37]	Arbor Acres	42	200	Standard
Salah et al. [10]	Ross 308	42	100	Standard
Salah et al. [38]	Ross 308	42	100	Standard
Shan et al. [39]	Not reported	52	200	Standard
Xie et al. [11]	Ross 308	49	500, 1000, 2000	Standard
Xu et al. [12]	Not reported	35	10	Standard
Yadav et al. [40]	Cobb 500	20	100, 200	Standard
Zhang et al. [41]	Arbor Acres	21	50, 100, 200	Standard
Zhang et al. [42]	Arbor Acres	42	50, 100, 200	Standard
Zhang et al. [43]	Arbor Acres	21	50, 100, 200	Standard
Zhang et al. [44]	Arbor Acres	21	50, 100, 200	Standard
Zhang et al. [45]	Arbor Acres	42	150	Standard

**Table 2 antioxidants-14-00460-t002:** Growth performance of broiler chickens supplemented with curcumin.

Item	N (NC)	Control Means (SD)	WMD (95% CI)	*p*-Value	Heterogeneity		Begg Test ^2^	Egger Test ^3^
					I^2^ (%)	*p*-Value ^1^	*p*-Value	*p*-Value
ADFI, g/d	22 (52)	103.92 (27.12)	0.333 (−0.678; 1.343)	0.519	82.95	<0.001	0.548	0.176
DWG, g/d	21 (51)	55.76 (13.05)	2.55 (1.250; 3.849)	<0.001	98.29	<0.001	0.572	0.079
FCR, g/g	20 (49)	1.86 (0.23)	−0.086 (−0.128; −0.043)	<0.001	97.77	<0.001	0.468	0.399

N: number of studies; NC: number of comparisons between treatments supplemented with curcumin and control; SD: standard deviation; WMD: weighted mean differences between treatments supplemented with curcumin and control; CI: confidence interval of WMD; I^2^: proportion of total variation in size effect estimates due to heterogeneity; ^1^ *p*-value to Cochran’s Q statistic; ^2^: Begg’s adjusted rank correlation; ^3^: Egger’s regression asymmetry test; ADFI: average daily feed intake; DWG: daily weight gain; FCR: feed conversion ratio.

**Table 3 antioxidants-14-00460-t003:** Antioxidant status in blood serum of broiler chickens supplemented with curcumin.

Item	N (NC)	Control Means (SD)	WMD (95% CI)	*p*-Value	Heterogeneity		Begg Test ^2^	Egger Test ^3^
					I^2^ (%)	*p*-Value ^1^	*p*-Value	*p*-Value
SOD, U/mL	7 (16)	232.95 (68.65)	29.043 (11.710; 46.376)	0.001	99.28	<0.001	0.608	0.639
CAT, U/mL	5 (9)	76.90 (18.95)	10.728 (3.056; 18.401)	0.006	92.76	<0.001	0.262	0.581
GSH-Px, U/mL	6 (15)	127.40 (37.52)	31.605 (22.022; 41.187)	<0.001	91.33	<0.001	0.381	0.877
T-AOC, U/mL	4 (4)	1.01 (0.33)	0.088 (0.038; 0.139)	<0.001	40.16	0.102	0.276	0.279
MDA, nmol/mL	8 (24)	1.87 (0.60)	−0.290 (−0.375; −0.206)	<0.001	47.84	0.072	0.126	0.535

N: number of studies; NC: number of comparisons between treatments supplemented with curcumin and control; SD: standard deviation; WMD: weighted mean differences between treatments supplemented with curcumin and control; CI: confidence interval of WMD; I^2^: proportion of total variation in size effect estimates due to heterogeneity; ^1^ *p*-value to Cochran’s Q statistic; ^2^: Begg’s adjusted rank correlation; ^3^: Egger’s regression asymmetry test; SOD: superoxide dismutase; CAT: catalase; GSH-Px: glutathione peroxidase; T-AOC: total antioxidant capacity; MDA: malondialdehyde.

**Table 4 antioxidants-14-00460-t004:** Intestinal morphology of broiler chickens supplemented with curcumin.

Item	N (NC)	Control Means (SD)	WMD (95% CI)	*p*-Value	Heterogeneity		Begg Test ^2^	Egger Test ^3^
					I^2^ (%)	*p*-Value ^1^	*p*-Value	*p*-Value
VH, µm	6 (18)	1268.29 (203.94)	236.891 (164.338; 309.443)	<0.001	89.51	<0.001	0.166	0.260
CD, µm	6 (18)	218.66 (35.12)	−17.233 (−26.113; −8.354)	<0.001	85.76	<0.001	0.102	0.519
VH/CD ratio	6 (18)	6.37 (1.14)	1.591 (1.023; 2.159)	<0.001	85.53	<0.001	0.578	0.335

N: number of studies; NC: number of comparisons between treatments supplemented with curcumin and control; SD: standard deviation; WMD: weighted mean differences between treatments supplemented with curcumin and control; CI: confidence interval of WMD; I^2^: proportion of total variation in size effect estimates due to heterogeneity; ^1^ *p*-value to Cochran’s Q statistic; ^2^: Begg’s adjusted rank correlation; ^3^: Egger’s regression asymmetry test; SOD: superoxide dismutase; CAT: catalase; GSH-Px: glutathione peroxidase; T-AOC: total antioxidant capacity; MDA: malondialdehyde; VH: villus height; CD: crypt depth.

**Table 5 antioxidants-14-00460-t005:** Meat quality of broiler chickens supplemented with curcumin.

Item	N (NC)	Control Means (SD)	WMD (95% CI)	*p*-Value	Heterogeneity		Begg Test ^2^	Egger Test ^3^
					I^2^ (%)	*p*-Value ^1^	*p*-Value	*p*-Value
Carcass yield, %	5 (16)	72.47 (2.83)	1.043 (0.389; 1.697)	0.002	89.92	<0.001	0.296	0.193
Meat pH	7 (18)	6.04 (0.32)	−0.089 (−0.196; 0.019)	0.106	39.61	0.303	0.606	0.852
Lightness (L*)	7 (17)	50.10 (5.79)	0.447 (−0.028; 0.922)	0.065	37.90	0.245	0.761	0.428
Redness (a*)	7 (17)	4.57 (1.23)	0.356 (0.095; 0.617)	0.007	47.14	0.150	0.953	0.713
Yellowness (b*)	7 (17)	5.07 (1.06)	0.393 (0.020; 0.767)	0.039	35.71	0.074	0.600	0.153
CL, %	3 (6)	14.23 (2.75)	−0.996 (−1.663; −0.330)	0.003	42.33	0.210	0.206	0.622
ShF, kgf/cm^2^	3 (9)	1.39 (0.11)	0.013 (−0.026; 0.052)	0.518	0.00	0.998	0.113	0.104
Meat composition g/100 g								
Fat	3 (9)	2.28 (0.67)	−0.256 (−0.361; −0.152)	<0.001	91.03	<0.001	0.482	0.439
Protein	3 (9)	22.08 (1.04)	0.261 (−0.253; 0.776)	0.319	41.83	0.442	0.267	0.241
MDA, mg/kg	3 (5)	1.13 (0.56)	−0.253 (−0.390; −0.117)	<0.001	88.40	<0.001	0.247	0.405

N: number of studies; NC: number of comparisons between treatments supplemented with curcumin and control; SD: standard deviation; WMD: weighted mean differences between treatments supplemented with curcumin and control; CI: confidence interval of WMD; I^2^: proportion of total variation in size effect estimates due to heterogeneity; ^1^ *p*-value to Cochran’s Q statistic; ^2^: Begg’s adjusted rank correlation; ^3^: Egger’s regression asymmetry test; CL: cooking loss; ShF: shear force; MDA: malondialdehyde.

**Table 6 antioxidants-14-00460-t006:** Meta-regression comparing the associations between covariates and measured outcomes.

Outcomes	Covariates	QM	Df	*p*-Value	R^2^ (%)
Average daily feed intake (ADFI)	Breed	0.596	3	0.378	1.56
	Supplementation period	6.092	1	0.014	11.45
	Curcumin dose	3.702	2	0.157	0.00
	Curcumin type	1.775	1	0.510	1.83
Daily weight gain (DWG)	Breed	2.829	3	0.816	0.71
	Supplementation period	6.482	1	0.011	12.04
	Curcumin dose	7.081	2	0.029	10.79
	Curcumin type	0.160	1	0.198	1.93
Feed conversion ratio (FCR)	Breed	2.912	3	0.491	0.81
	Supplementation period	0.365	1	0.546	0.00
	Curcumin dose	6.273	2	0.043	9.16
	Curcumin type	0.135	1	0.714	0.00

QM: coefficient of moderators; QM is considered significant at *p ≤* 0.05; Df: degree of freedom; R^2^: the amount of heterogeneity accounted for.

## Data Availability

The datasets used and analyzed during the current study are available from the corresponding author on reasonable request. The data are not publicly available due to restrictions on privacy.

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
