# Peer review of "Meta-Analysis of Dietary Curcumin Supplementation in Broiler Chickens: Growth Performance, Antioxidant Status, Intestinal Morphology, and Meat Quality"

_antioxidants, 2025, doi:10.3390/antiox14040460_

Round 1
Reviewer 1 Report
The regulatory mechanisms of curcumin on intestinal health could be further supplemented in the discussion section to enhance the readability and comprehensiveness of the article
The regulatory mechanisms of curcumin on intestinal health could be further supplemented in the discussion section to enhance the readability and comprehensiveness of the article
Author Response
Response to Reviewer #1:
We would like to thank the reviewer for the careful and thorough reading of this manuscript and for the thoughtful comments and constructive suggestions, which help to improve the quality of this manuscript. Our response follows (the reviewer's comments are in italics).
Comments and Suggestions for Authors
Comment 1. The regulatory mechanisms of curcumin on intestinal health could be further supplemented in the discussion section to enhance the readability and comprehensiveness of the article.
Response 1: Dear reviewer, it is impossible to add further explanations on the regulatory mechanisms of intestinal health in the discussion section because the 28 scientific articles included in the current study database did not evaluate curcumin's mechanisms of action other than those presented in the discussion section of our manuscript. Likewise, we did not find additional information on curcumin's mechanisms of action on intestinal health in other species because curcumin has primarily been evaluated in broiler chickens. Therefore, adding more information on intestinal health, as you suggest, is impossible.
Reviewer 2 Report
The authors have performed a systemic review to determine the impact of curcumin on the production, performance, antioxidant status and meat quality in broiler chickens. It is an important contribution to the field of poultry nutrition, particularly in its potential to improve feed conversion ratio, meat quality and antioxidant status. This manuscript would serve as a valuable resource in the field of broiler nutrition.
The manuscript was written very well with appropriate scientific content by including an introduction, methods, results with appropriate figures, discussion and conclusion.
Author Response
Response to Reviewer #2:
We would like to thank the reviewer for carefully and thoroughly reading this manuscript and for accepting our manuscript in its current form without further corrections.
Reviewer 3 Report
The present paper is a relevant contribution and will increase the knowledge in the studied field. The paper can be accepted in present form.
The present paper is a relevant contribution and will increase the knowledge in the studied field. The paper can be accepted in present form.
Author Response
Response to Reviewer #3:
We want to thank the reviewer for their careful and thorough reading of this manuscript and for accepting our manuscript in its current form.